# Pharmaceutical Potential of Remedial Plants and Helminths for Treating Inflammatory Bowel Disease

**DOI:** 10.3390/ph17070819

**Published:** 2024-06-21

**Authors:** Tenzin Jamtsho, Alex Loukas, Phurpa Wangchuk

**Affiliations:** 1College of Public Health, Medical, and Veterinary Sciences (CPHMVS), Cairns Campus, James Cook University, Cairns, QLD 4878, Australia; 2Australian Institute of Tropical Health and Medicine (AITHM), Cairns Campus, James Cook University, Cairns, QLD 4878, Australia; alex.loukas@jcu.edu.au

**Keywords:** natural product, small molecule, cytokines, remedial plants, helminths

## Abstract

Research is increasingly revealing that inflammation significantly contributes to various diseases, particularly inflammatory bowel disease (IBD). IBD is a major medical challenge due to its chronic nature, affecting at least one in a thousand individuals in many Western countries, with rising incidence in developing nations. Historically, indigenous people have used natural products to treat ailments, including IBD. Ethnobotanically guided studies have shown that plant-derived extracts and compounds effectively modulate immune responses and reduce inflammation. Similarly, helminths and their products offer unique mechanisms to modulate host immunity and alleviate inflammatory responses. This review explored the pharmaceutical potential of Aboriginal remedial plants and helminths for treating IBD, emphasizing recent advances in discovering anti-inflammatory small-molecule drug leads. The literature from Scopus, MEDLINE Ovid, PubMed, Google Scholar, and Web of Science was retrieved using keywords such as natural product, small molecule, cytokines, remedial plants, and helminths. This review identified 55 important Aboriginal medicinal plants and 9 helminth species that have been studied for their anti-inflammatory properties using animal models and in vitro cell assays. For example, curcumin, berberine, and triptolide, which have been isolated from plants; and the excretory-secretory products and their protein, which have been collected from helminths, have demonstrated anti-inflammatory activity with lower toxicity and fewer side effects. High-throughput screening, molecular docking, artificial intelligence, and machine learning have been engaged in compound identification, while clustered regularly interspaced short palindromic repeats (CRISPR) gene editing and RNA sequencing have been employed to understand molecular interactions and regulations. While there is potential for pharmaceutical application of Aboriginal medicinal plants and gastrointestinal parasites in treating IBD, there is an urgent need to qualify these plant and helminth therapies through reproducible clinical and mechanistic studies.

## 1. Introduction

Inflammation is a defense response to injuries, chemicals, or microorganisms, involving by tissue damage, as well as metabolic and microcirculation disorders. It involves the release of mediators such as nitric oxide, prostaglandins, and cytokines, which are essential for removing harmful stimuli, restoring normal physiology, and controlling inflammation [1]. Different cytokines, markedly, interleukin-1 (IL-1) and tumor necrosis factor (TNF), regulate inflammation. These chemical mediators, primarily triggered by bacterial lipopolysaccharide [2], are released by monocytes, macrophages, and other cells [3]. Anti-inflammatory drugs have been used to address severe inflammation and pain. Presently, the United States Food and Drug Administration (USFDA) has approved nonsteroidal anti-inflammatory drugs (NSAIDs), including indomethacin, ibuprofen, cyclooxygenase-2 enzyme (COX-2) inhibitors like celecoxib and naproxen, for managing inflammation. However, these NSAIDs are linked to side effects on various bodily systems [4]. For instance, the long-term use of NSAIDs increases the risk of ulcers and complications in the upper gastrointestinal tract by about four times, while in the lower gastrointestinal tract, they cause mucosal ulcers or erosions and alter the diversity of microbiota, thereby modulating their toxicity [5].

Among the various inflammatory disorders, the chronic-relapsing condition of IBD, which mainly affects the gastrointestinal tract, has become a global health burden [5]. The etiology and pathogenesis of IBD are multifaceted, and it is difficult to determine a single causative agent [6]. Overall, these disorders are marked by persistent inflammation, often resulting in complications such as a heightened likelihood of hospitalization, surgical intervention, colorectal cancer, and disability. Acknowledging the chronic and advancing characteristics of IBD, commencing effective treatment during the early phases of the disease is crucial to minimize relapses and prevent complications [7]. In this view, NSAIDs such as 5-aminosalicylic acids (5-ASA), corticosteroids, and immunomodulators such as 6-mercaptopurine and azathioprine are used for managing IBD [7]. These medications can induce and sustain remission [8]; unfortunately, they are linked to numerous side effects [9,10]. For instance, 5-ASA and corticosteroids, often prescribed for patients with mild-to-moderate ulcerative colitis, have exhibited only moderate efficacy. Long-term medication use is associated with side effects such as stunted growth, depression, hypertension, and osteoporosis [11].

Similarly, immunomodulators such as 6-mercaptopurine and azathioprine are also associated with pancreatic inflammation, decreased hematopoiesis, and liver toxicity [12]. Among the biological agents or biologics, infliximab is commonly utilized for IBD treatment. However, biologics are expensive and associated with adverse infusion lupus-like syndrome and infections, including sepsis [11,13]. Historically, natural products (NPs) have been a vital source of structurally diverse and pharmacologically important anti-inflammatory small molecules (SMs, <10 kDa molecular weight) [14,15,16,17,18]. For instance, in the period spanning from 1981 to 2019, the USFDA approved 33.5% (1394 SMs) of the small-molecule (SM) drugs derived from natural products, underscoring the significant role of NPs in shaping the global healthcare landscape [18]. A total of 1881 new drugs were approved, which included 71 drugs sourced from unaltered natural products and 14 from botanicals. Among the 1602 new chemical entities, 53 were identified to exhibit anti-inflammatory properties [18].

Overall, the treatment of inflammatory disorders such as IBD has seen a transition from focusing on symptomatic control to emphasizing more objective endpoints [19]. This has resulted in the development of large-molecule biologics and synthetic small-molecule drugs (SMDs). However, these drugs have efficacy, safety, cost, and management time frame limitations. Considering this, more cost-effective oral drugs that offer improved efficacy and tolerability are in demand. Consequently, there is rising interest in developing new, targeted anti-inflammatory SMDs for IBD and other immune-mediated inflammatory conditions. Due to their low molecular weight, SMDs can quickly diffuse through cell membranes. The variance in size notably influences aspects such as the route of administration, target site, pharmacokinetics, antigenicity, and drug interactions. When administered orally, these drugs resist gastric degradation and quickly enter into the systemic circulation. The short half-life of SMDs can prove advantageous, especially in settings where instant drug elimination occurs, such as in infections, during surgical procedures, or during pregnancy. Hence, herbal and natural product compounds have been screened for therapeutic potential in anti-inflammatory-related diseases due to their chemical diversity, lower toxicity, and cost-effectiveness. However, biocompatibility and toxicity concerns have impeded clinical trial progress [20]. To address issues like poor bioavailability, solubility, and targeted delivery, advanced drug delivery systems such as nanoparticles, bioadhesive microspheres, chitosan-based hydrogels, liposomes, and phytosomes are being developed [21,22,23,24,25,26]. Nanoparticles can encapsulate or attach therapeutic drugs, ensuring precise targeting and controlled release, enhancing drug delivery and efficacy. FDA-approved nanoparticle-based therapies, such as liposomes and micelles, improve drug delivery by preventing gastrointestinal degradation and enhancing the delivery of poorly soluble drugs [27,28,29]. Similarly, the phytophospholipid complex technique converts water-soluble herbal extracts into lipid-compatible complexes, improving absorption and protecting compounds from degradation, thus enhancing therapeutic efficacy [30]. Examples of nanoformulated phytochemicals include silymarin, hypericin, and curcumin [31].

While general anti-inflammatory drugs have been reviewed elsewhere in more detail, we examined emerging the SMDs with anti-inflammatory properties derived from natural sources for treating IBD. Specifically, we explored anti-inflammatory SMs in medicinal plants and gastrointestinal parasites, offering insights into their potential future roles and impact. Databases such as Scopus, MEDLINE Ovid, PubMed, Google Scholar and Web of Science were queried using keywords like ‘Aboriginal remedial plants’, ‘helminths therapy’, and ‘anti-inflammatory small molecules’ to uncover the literature discussing the biological activities, phytochemistry, and ethnomedical applications of both Aboriginal plants and helminths against inflammatory-related disorders.

## 2. Plant-Derived Anti-Inflammatory Compounds

Plants have been a healing source since time immemorial [32]. Since their chemical analysis in the 19th century, bioactive compounds, referred to as phytochemicals in plants, animals, and living organisms, which influence physiological processes and have therapeutic potential and offer health benefits [33,34], have been crucial in advancing drug development [35]. Nearly 150,000 plant species, including medicinal plants, have been studied, with numerous harboring valuable therapeutic compounds, particularly anti-inflammatory compounds [36,37]. For example, the *Eucalyptus* and *Melaleuca* species are herbal remedies used by Australian Aboriginal people to treat inflammatory ailments such as muscular aches, sores, internal pains, and painful joints [38]. Likewise, Bhutanese traditional medicine involves the use of *Aconitum laciniatum* Stapf and the entire plant of *Aconitum orochryseum* Stapf to treat chronic parasitic and microbial infections, inflammatory diseases, and bilious fever [39]. Considering these traditional practices, researchers isolated the compound 14-O-acetylneoline, which exhibited anticolitis activity in TNBS (2,4,6-trinitrobenzene sulfonic acid)-induced colitis [40,41]. This practice emphasizes the long-standing recognition of the anti-inflammatory properties of NPs. Patients with IBD often turn to botanicals due to their perceived safety and effectiveness. Popular herbal remedies include *Tripterygium wilfordii* Hook. f., *Plantago ovata* Phil., *Artemisia absinthium* L., *Aloe vera* L., *Curcuma longa* L., *Boswellia serrata* Roxb., and *Cannabis sativa* L. [42]. A review of 27 studies and 1874 patients with IBD found seven herbal remedies were beneficial for ulcerative colitis (UC) and four induced remissions in cystic fibrosis diseases [42]. The oral administration of *Aloe vera* gel and *Plantago ovata* Phil. seeds showed promising anti-inflammatory results in UC [42]. Overall, ethnopharmacology insight has been a guiding force in the biodiscovery of anti-inflammatory SMs from NPs.

Inflammatory mediators play a crucial role in modulating IBD through various mechanisms. For example, nitric oxide (NO) is a signaling molecule produced by inducible nitric oxide synthase (iNOS) during inflammation and plays a significant role in vasodilation, modulation of blood flow, and immune defense. However, excess NO production can contribute to inflammation and tissue damage [43]. Both crude extracts and compounds isolated from natural sources exhibit anti-inflammatory effects by regulating key cytokines [44,45]. Pro-inflammatory cytokines like IL-6 promote the immune response by stimulating acute-phase protein production and influencing B-cell differentiation, while TNF and IL-1β, produced mainly by macrophages, mediate inflammation by promoting leukocyte recruitment, fever, and apoptosis in infected cells. Conversely, anti-inflammatory cytokines such as IL-10, IL-4, and IL-13, released by T helper 2 (Th2) cells, inhibit pro-inflammatory responses, modulating immune functions and inflammation [46]. However, excess cytokines are associated with chronic inflammation and autoimmune diseases.

Both crude extracts and compounds isolated from medicinal plant species have demonstrated wide-spectrum anti-inflammatory activity in both in vitro and in vivo screening (Table 1). For example, a methanol extract of *Blainvillea acmella* (L.) Philipson inhibited interleukin (IL)-1β and IL-6 expression levels in LPS-stimulated RAW264.7 macrophage cells [47]. This bioactivity guided the isolation of the bioactive compound spilanthol from *B. acmella*, inhibiting iNOS expression NO production and reducing TNF levels in LPS- and IFNγ-induced RAW264.7 cells [48]. In another study, a methanol extract of *Corymbia terminalis* (F.Muell.) K.D.Hill and L.A.S.Johnson inhibited IL-6, IL-8, and COX-1 expression levels in LPS-stimulated mammalian keratinocyte (HaCaT) cells [49,50]. Small molecules such as axifolin, aromadendrin, cianidanol, and farrerol isolated from *C. terminalis* showed anti-inflammatory properties by suppressing IL-6, IL-8, and cyclooxygenase-1 (COX-1) expression in LPS-stimulated HaCaT cells [49] (Table 1). Likewise, an ethanol root extract of *Euphorbia tirucalli* L. inhibited the TNF and interferon-gamma (IFN-γ) production in carrageenan-induced acute inflammation in the albino rat hind paw edema model [51]. Phytochemically, *Eucalyptus* species were reported to contain 1,8-cineole and α-pinene and aromadenderene as the main compounds in their leaves and essential oils [52]. The compound 1,8-cineole demonstrated the downregulation of inflammatory responses in dextran sulfate sodium (DSS)-induced colitis in mice and decreased proinflammatory chemokine production in TNF-stimulated HT-29 cells, proving to be a potent compound in treating human IBD [53]. Monoterpene acid and gallic acid glucose esters and phenolic compounds have also been reported in *Eucalyptus* species, which have shown a substantial reduction in the production of pro-inflammatory cytokines, including TNF, IL-1β, and IL-6 in LPS-stimulated peripheral blood mononuclear cells (PBMCs) and LPS-stimulated murine macrophage (RAW264.7) cells, indicating significant anti-inflammatory effects [54,55]. Studies have also reported the presence of phenolic compounds (including flavonoids, phenylpropanoids, and polyphenols) [56,57,58] and triterpenoids (including lupine, ursane, and oleanane derivatives) in *Melaleuca* species [59,60,61,62]. Many of these compounds showed pharmacological activities, including anti-inflammatory properties [63]. For instance, galloyl-lawsoniaside A and (4S)-α-terpineol 8-O-β-D-(6-O-galloyl) glucopyranoside, isolated from *Uromyrtus metrosideros* (F.M.Bailey) A.J.Scott, demonstrated remarkable in vitro inhibition of proinflammatory cytokines, which are linked to the development of IBD. Specifically, the releases of IFN-γ, IL-17A, and IL-8 from phorbol myristate acetate/ionomycin (P/I) and anti-CD3/anti-CD28-activated T cells were significantly suppressed [64]. On a similar note, the isolated compound hispidulin from *Clerodendrum inerme* R.Br. exhibited anti-inflammatory properties characterized by the inhibition of prostaglandin E2 (PGE2) production. It suppressed the expression of inducible nitric oxide synthase (iNOS) and COX-2 by blocking nuclear factor-kappa B (NF-κB) DNA-binding activity in LPS-stimulated RAW264.7 macrophages [2].

Many researchers have conducted studies to assess the effectiveness of plant-derived extracts and compounds in chronic IBD models. These have included curcumin (1E,6E)-1,7-bis (4-hy- droxy-3-methoxyphenyl) hepta-1,6-diene-3,5-dione, isolated from *Curcuma longa* L. (Zingiberaceae) [65]; colchicine, a major alkaloid from *Colchicum autumnale* L. [66]; resveratrol from *Veratrum grandiflorum* O.Loes. [67]; capsaicin from *Capsicum species* [68]; and epigallocatechin-3-gallate (EGCG) from *Camellia sinensis* (L.) Kuntze [69]. Additional small bioactive molecules, such as quercetin and berberine, have been shown to suppress IFN-γ- and IL-17A; while berberrubine, which inhibits myeloperoxidase (MPO), has been identified and isolated from *Berberis vulgaris* [70,71,72]. In clinical studies, curcumin inhibits important proinflammatory signaling cascades, such as NF-*κ*B, mitogen-activated protein kinases (MAPK), COX and lipoxygenase (LOX) pathways in LPS-stimulated RAW264.7 cells and HT-29 human colon cancer cells. Additional proinflammatory cytokines like TNF, IL-1β, and IL-6 were downregulated [73,74,75]. According to numerous studies, curcumins exhibit a good safety profile (well tolerated and nontoxic).

Other anti-inflammatory lead compounds such as brevilin A, centiplide A and H (*Centipeda minima*) [76,77], geraniin, corilagin (*Acalypha wilkesiana* Mull.Arg.) [78], 5′-methoxy nobiletin, 1,2-benzopyrone, (*Ageratum conyzoides* L.) [79], costatamins A–C (*Angophora costata* Britten), quercetin 3-O-(2″-acetyl)-glucoside, oumarinolignoid, cleomiscosins A–C (*Arivela viscosa* (L.) Raf.) [80,81], and quercetin 7-O-β-D-glucopyranoside (*Brasenia schreberi* J.F. Gmel.) have been isolated following ethnopharmacology and bioassay-guided approaches. The quercetin and quercitrin isolated from *Merremia tridentata* (L.) Hallier f. exhibited inhibitory effects on NO production and proinflammatory cytokines such as IL-6, TNF, and IL-1β in LPS-stimulated RAW264.7 cells [82]. Apigenin demonstrated similar bioactivity in LPS-stimulated BV2 microglia [83]. Additionally, calophyllolide and 27-[(E)-p-coumaroyl] canophyllic derived from *Calophyllum inophyllum* L. downregulated IL-6, TNF, IL-1β, and NO production while upregulating IL-10 in LPS-stimulated RAW 264.7 cells [4]. Compounds including antidesoside, podocarpusflavone A, and amentoflavone from *Antidesma bunius* Wall. also reduced NO levels in LPS-stimulated BV2 cells and RAW 264.7 cells [4]. The crude extracts and a few isolated compounds from NPs showing anti-inflammatory activity in diverse colitis animal models are summarized in Table 1. Representative chemical structures are shown in Figure 1.

**Table 1 pharmaceuticals-17-00819-t001:** Anti-inflammatory activity of crude extract and bioassay-guided isolated SMs from Aboriginal medicinal plants.

Botanical Name and Family	Plant Parts and Traditional Uses	Anti-Inflammatory Compounds/Extracts.	Model/Cell Used for Testing	Main Effect on Inflammation	Ref
*Abrus precatorius* L. (Fabaceae)	Seeds; abortion	Olean-12-ene-3β, 16β,23,28-tetrahydroxy-3-O-{[β-D-glucopyranosyl-(→ 4)-β-D-glucopyranosyl-( 1→ 3)]-[β-D-glucopyranosyl-(→ 2)]-β-D-fucopyranose}	Croton oil ear model	↓ Inflammation on the ear tissue of rats	[4]
		(S)-8-Hydroxy 6,7, 5′-trimethoxyisoflavan-1′,4′-quino Abruquinone A and B	Polymorphonuclear cells (PMNs)	↓ TNF and ROS	[84]
*Acacia melanoxylon* R.Br. (Fabaceae)	Bark: headache, cold and fever	Kolavic acid 15-methyl ester	LPS- LPS-stimulated J774 cell	↓IL-6	[85]
*Acalypha wilkesiana* Müll.Arg. (Euphorbiaceae)	Shoot: sores/skin lesions/wounds/cuts	Polyphenol-enriched fraction	LPS-stimulated RAW264.7 macrophage	↓ TNF, IL- 1β, and IL-6	[78,86]
*Ageratum conyzoides* L. (Asteraceae)	Whole plant: sores/skin lesions/wounds/cuts	5′-Methoxy nobiletin 1, 2-benzopyrone	Carrageenan-induced pleurisy	↓ p-p65 NF-κB and p-p38 MAPK ↑ IL-17A, IL-6, TNF and IFN-γ levels	[87,88]
		Leaves extract and aerial extract	Cotton pellet-induced granuloma and formaldehyde-induced arthritis models	↓ Paw edema	
*Alphitonia petriei* Braid and C.T. White (Rhamnaceae)	Bark, leaves, stem; Body pain	Embolic acid	LPS + IFN-γ activated RAW 264.7 cells	↓ NO and TNF production	[89]
*Alphitonia excelsa* Reissek ex Endl (Rhamnaceae).	Whole plants: headache, colds, fever, stomach upset, skin lesions, wounds, cuts	Betulinic acid	λ-carrageenan-induced paw edema mice	↓ COX-2, NO, TNF, and IL1-β	[90]
*Angophora costata* Britten (Myrtaceae)		Costatamins A-C	LPS-stimulated RAW264.7 cell	↓ NO production and TNF	[91]
*Alstonia scholaris* (L.) R.Br. (Apocynaceae)	Juice, sap, bark; toothache, fever, sores, skin lesions, wounds, and cuts	12-ursene-2,3,18,19-tetrol-28 Acetate	Carrageenan-induced. paw edema (Wistar rats)	↓ Paw edema	[92]
		Picrinine, vallesamine, and scholaricine	Mice air pouch model (In vivo)	↑ SOD activity ↓ NO production, PGE2, and MD	[93]
*Antidesma bunius* Wall. (Phyllanthaceae)	Fruit: colds, fever, and headache	Antidesoside, podocarpusflavone A, byzantionoside B, (6S,9R)-roseoside	LPS-stimulated BV2 cells and RAW264.7 macrophages	↓ NO production	[94]
*Arivela viscosa* (L.) Raf. (Cleomaceae)	Whole plants; Colds and fever	Quercetin 3-O-(2″-acetyl)-glucoside	Carrageenan-induced rat paw edema	↓ Carrageenan-induced rat paw edema	[80]
		Coumarinolignoid cleomiscosins A, B and C	Female Swiss albino mice	↓ IL-6, TNF NO production ↑ IL-4 in a dose-dependent manner	[81]
				↓ Carrageenan-induced rat paw edema ↓ IL-4, TNF, and NO production ↓ COX-1	
		Cembrenoid diterpene Malabaric acid	Cyclooxygenase enzyme (COX-1 and -2) inhibitory assays	↓ COX-1 and COX-2 enzyme	[95]
		Stigmast-4-ene-3,6-dione, cleomaldeic acid, stigmast-4-en-3-one, and lupeol		↓ COX-1 and COX-2 enzyme	
*Barringtonia racemose* (L.) Spreng. (Lecythidaceae)	Bark, fruits; tonic, pain and inflammatory	Ethyl acetate fraction	Carrageenan-induced rat paw edema model	Demonstrated dose-dependent anti-inflammatory activity	[96]
		Barringosides I	LPS-stimulated RAW264.7 cell	↓ Production of NO	[97]
*Blainvillea acmella* (L.) Philipson (Asteraceae)	Bark, fruits, and roots; muscle sprain, bone aches, dislocation, broken bones	Spilanthol	LPS and IFNγ induced RAW264.7 cells	↓ iNOS expression, NO production, and TFN	[48]
		Methanol extracts	LPS-stimulated RAW264.7 macrophages	↓ IL-1β and IL-6	[47]
*Boerhavia diffusa* L. (Nyctaginaceae)	Whole plants; asthma	Boeravinone B and N	Carrageenan-induced paw edema	↓ COX-1 and COX-2 Exhibited anti-inflammatory activity	[98]
		Rotenoid-rich fraction	Sprague–Dawley rats	Exhibited ↑ anti-inflammatory potential and ↑ plasma level	[99]
*Brasenia schreberi* J.F.Gmel. (Cabombaceae)	Leaves: stomach cancer, boil, dysentery, and tuberculosis	Quercetin 7-0-β-d-glucopyranoside	LPS-stimulated RAW 264.7 cells	↓ Expression of iNOS and NO Overexpression of COX-2 ↓ Granulocyte macrophage-colony-stimulating factor	[100]
*Brucea javanica* (L.) Merr. (Simaroubaceae)	Leaves and roots; pain	Ethyl acetate fraction of seeds	LPS-stimulated RAW264.7 macrophages	↓ NO, PGE2, TNF, IL-1β, and IL-6 ↑ IL-10	[101]
			Carrageenan-induced paw edema	Inhibited carrageenan-induced paw edema	
		Oil emulsion	DSS-induced colitis mice	↓ TNF, IL-1β, IL-6, IL-8, IL-17, and IFN-γ ↓ mRNA expression of MPO, iNOS, and COX-2	[102]
		Cleomiscosin A and E	LPS-stimulated RAW264.7 macrophages	↓ NO production	[103]
		Brujavanoid E	LPS-induced RAW264.7 cells	↓ Caspase1, CD206, IL-1β, IL-18, MCP-1, TNF, NIK, NLRP3, and p65 in a dose-dependent manner	[104]
		Brusatol	LPS-induced RAW 264.7 cells TNBS-induced colitis mice	↓ TNF, IL-1β, PGE2, and NO levels ↓ NF-κB signaling pathway. ↓ IL-1β and IL-18 levels ↓ Catalase, glutathione, and superoxide dismutase enzymes in the colon tissue	[105,106,107]
*Calophyllum inophyllum* L. (Clusiaceae)	Fruits; body pain and purgative	Acetone extract	LPS-stimulated RAW 264.7 cells	↓ NO production, iNOS, COX-2, and NF-κB	[108]
		Calophyllolide	Albino male mice	↓ IL-1β, IL-6, TNF ↑IL-10	[109]
		E-coumaroyl triterpenoid (27-[(E)-p-coumaroyloxy] canophyllic acid)	LPS-induced RAW 264.7 cells	↓ NO production, IL-1β, and TNF, iNOS ↓ NF-κB signaling pathway	[110]
*Capparis mitchellii* Lindl. (Capparaceae)	Bark: cuts, wounds, and skin lesions	Luteolin, kaempferol, apigenin and catechin	LPS-induced RAW 264.7 cells	Inhibition with IC50 values of 26.24 ± 1.78, 20.38 ± 1.36, 22.8 ± 1.57, and 42.5 ± 2.24 μM, respectively ↓ NO production	[111]
*Salvia plebeian* R.Br. (Labiatae)	Colds and tumors	Ethanol extract	BALB/c mice	↓ IL-4, IL-17, MMP-1, and MMP-3 ↓ Akt and MAPKs pathways, ↓ Akt and ERK p38 expression	[112]
*Carpobrotus rossii* Schwantes (Aizoaceae)	Leaves: (extract juice) respiratory tract and throat infections, gastrointestinal discomfort, insect bites, wounds burns, eczema, bluebottle, and jellyfish stings	Aqueous extract	PBMC	↓ IL-10, TNF, and MCP-1	[113]
*Clematis pickeringii* A. Gray (Ranunculaceae)	Headache, pain, rheumatism, infections, common colds.	Ethanolic extract	HepG2 cells	↑ PPARa and PPARy at the dose of 60 µg/mL	[114]
*Centipeda minima* (L.) A.Br. (Apiaceae)	Nasal allergy, diarrhea, asthma	Whole-plant extract	LPS-induced RAW 264.7 cells and λ-carrageenan-induced paw edema	↓ NO production	[115]
			DSS-induced colitis mouse model	↓ TNF-α and IL-1β	[116,117]
		Brevilin A, centiplide A, centiplide H, and Helenalin isova lerate	LPS-induced RAW 264.7 cells	Attenuated NF-κB pathway activation and oxidative stress ↓ NO production	[76,77]
		6-O-angeloylplenolin	LPS-induced macrophage cells (BV2 cells)	↓ TNF and IL-1β ↓ Phosphorylation of NF-κB ↓ IκB-α, NO, and PGE2	[76,118]
*Centella asiatica* (L.) Urb. (Apiaceae)	Whole plants; lupus, varicose ulcers, eczema, and psoriasis	Triterpenoid saponin-rich fraction	LPS-stimulated RAW 264.7 cells	↓ IL-13, NF-κB pathway	[119]
		Asiatic acid, isomadecassoside, asiaticoside B and G, rosmarinic acid	LPS-stimulated RAW 264.7 cells	↓ NO production	[120,121]
*Clematis microphylla* DC. (Ranunculaceae)	Whole plants; headache, colds, sores, gastric disorder and fever	Ethanolic plant extracts	HepG2 cells	↓ COX-1, 5-LOX	[114]
*Clerodendrum inerme* R.Br. (Lamiaceae)	Leaves and roots; sores, skin lesions, wounds, cuts, and sprains	Ethanol leaf extract	LPS-induced RAW 264.7 cells	↓ NO production ↓ mRNA and protein expressions of iNOS	[122]
		Hispidulin	LPS-stimulated RAW 264.7 cells	↓ NO production, NF-κB DNA-binding activity, and JNK signaling pathway ↓ iNOS and COX-2 expression	
		Methanolic extract	Formalin-induced hind-paw edema	↑ Anti-inflammatory activity at dose 200 mg/kg	[123]
		Ethyl acetate fraction	LPS-stimulated RAW264.7 macrophage cells	↓ NO production ↓ iNOS	[122,124]
*Cleome viscosa* L. (Capparidaceae)	Whole plants; diarrhea, fever, cuts, and ulcers	Quercetin 3-O-(2″-acetyl)-glucoside	Carrageenan-induced paw edema	↓ Carrageenan-induced rat paw edema	[80,124]
		Cleomiscosins A-C	LPS-stimulated Peritoneal Macrophages (Swiss albino mice)	↓ IL-4, TNF, and NO production	[81]
		Malabaric acid		↓ COX-1 and two activities	[95]
		Hispidulin		↓ PGE2 production, and iNOS and COX-2 expressions ↓ NF-κB DNA-binding activity and JNK pathway	[122]
*Crinum pedunculatum* R.Br. (Amaryllidoideae)	Whole plants; stings from marine life	Methanol, ethanol, and ethyl acetate extracts	Carrageenin-induced Wistar albino paw edema	↓ Carrageenin-induced rat paw edema	[124,125]
*Morinda citrifolia* (Rubiaceae)	Fruits: cough and cold, sore throat		LPS-stimulated human monocyte	↓ Matrix metalloproteinase-9 (MMP-9) production	[126]
		Fruit juice	DSS-induced colitis model	↓ TNF and IFN-γ, NO and IL-17	[127]
		Ethyl acetate extract	Caco-2 cells	↓ COX-2, IL-8, and prostaglandin E2 production and neutrophil chemotaxis	[128]
		Fruit extract	LPS-stimulated RAW 264.7 cells	↓ NO synthase and TNF	[129]
		Rutin, nonioside A, (2E,4E,7Z)-deca-2,4,7-trienoate-2-O-β-d-glucopyranosyl-β-d-glucopyranoside, and tricetin	LPS-stimulated RAW 264.7 cells	↓ NO production, expression of IKKα/β, I-κBα, and NF-κB p65	[126,130]
*Eucalyptus camaldulensis* Dehnh. (Myrtaceae)	Leaves and gum from barks; diarrhea	Ethanol extract	Carrageenan-induced paw edema	↓ Carrageenan-induced paw edema	[124,131]
		12-O -tetradecanoyl-phorbol-13-acetate		↓ Edema and leukocyte infiltration	
*Acalypha wilkesiana* Mull.Arg. (Euphorbiaceae)	Shoots: skin lesions, sores, cuts, wounds	Leaf extract	LPS-stimulated RAW 264.7 cells	↓ NO and PGE2 ↓ iNOS productions and COX-2 expression ↓ TNF, IL-1β) and IL-6	[2]
*Corymbia terminalis* (F.Muell.) K.D.Hill and L.A.S.Johnson (Myrtaceae)	Bark; dysentery	Methanol leaves extracts	LPS- stimulated mammalian keratinocyte (HaCaT) cell	↓ IL-6, IL-8, and COX-1 and 2 enzyme activities	[49,50]
		Axifolin, aromadendrin, cianidanol, and farrerol		↓ IL-6, IL-8, and COX-1 and 2 enzyme activities	[49]
*Dodonaea polyandra* Merr. and L.M.Perry (Sapindaceae)	Roots: sores, skin lesions, wounds, cuts, and toothache	Nonpolar hexane and methylene chloride/methanol	TPA-induced mouse ear edema	↓ Inflammation in TPA-induced mouse ear edema by 72.12 and 79.81%, respectively, at 0.4 mg/ear; 12 and 79.81%, respectively, at 0.4 mg/ear	[132]
		Polyandric acid A	Mouse Ear Tissue (male BALB/c mice)	↓ IL-1β	[133]
		5,16-Epoxy-8α-(benzyloxy) methyl-2α-hydroxycleroda-3,13(16),14-tried-18-oic acid and 15,16-Epoxy-2α-benzoyloxycleroda-3,13(16),14-tried-18-oic acid	TPA-induced mouse ear edema	Exhibited optimum inhibition of inflammation (70–76%) at doses of 0.22 and 0.9 μmol/ear, respectively	[134]
*Dodonaea viscosa* Jacq. (Sapindaceae)	Leaves, rheumatism, skin infections, and diarrhea	Hydroalcoholic extract	Carrageenin-induced paw edema	↓ Edema induced in rats by carrageenin	[135]
		Dichloromethane extract	TPA-induced edema model in CD-1 mice	↓ 97.8% of the edema	[136]
		Hautriwaic acid	12-O-tetradecanoylphorbol 13-acetate (TPA) mice ear edema models	97% of edema inhibition with an ED50 = 0.158 mg/ear	
*Excoecaria agallocha* L. (Euphorbiaceae)	Latex; stings (marine)	Agallochanin K	RAW264.7 cells	↓ NF-κB	[137]
		Agallolides I and J	NF-κB (p65) RAW264.7 cells	↓ NF-κB with inhibition rates of 23.4% and 19.4%, respectively, at the concentration of 100.0 µM	[138]
*Eleocharis dulcis* Hensch. (Cyperaceae)	Whole plants; antiseptic, wounds	Ethyl acetate extract	LPS-induced RAW 264.7 macrophages	↓ TNF, iNOS, and COX-2	[124,139]
*Ipomoea pes-caprae* (L.) R.Br. (Convolvulaceae)	Whole plants; diseases, boils, and swelling	Ethanol extract from leaves and stems	Trypsin-, histamine-, and bradykinin-induced paw edema in mice	↓ Trypsin-, histamine-, and bradykinin-induced paw edema in mice	[140]
*Heliotropium ovalifolium* Forssk. (Boraginaceae)	Body wash and fever	4,7,8-Trimethoxy-naphthalene-2-carboxylic acid and 6-hydroxy-5,7-dimethoxy-naphthalene-2-carbaldehyde	LPS-stimulated THP-1 cell	↓ IL-6 and TNF	[141]
*Euphorbia hirta* L. (Euphorbiaceae)	Leaves; hypertension, respiratory ailments, tumors, and wounds	Ethanol extract	LPS-induced inflamed rat knees	↓ TNF and NO production	[142]
		Butanol extract	LPS-stimulated RAW 264.7 cells	↓ NO production and iNO protein expressions	[143]
*Melaleuca leucadendra* L. (Myrtaceae)	Leaves and bark; cough and cold; wounds, cuts, and sores	Butanol extract	LPS-stimulated RAW 264.7 cells	↓ NO and PGE2 production ↓ Expression of COX-2 and iNOS protein ↓ NF-κB transcriptional activity in a dose-dependent manner	[144]
		Casuarinin	Ethanol-induced gastric ulceration in rats	↓ NF-κB, COX-2, iNOS	[145]
*Euphorbia tirucalli* L. (Euphorbiaceae)	Latex; skin cancer	Ethanol extract from roots	Carrageenan-induced acute-inflammation in albino rat model	↓ TNF and IFN-γ production	[51]
*Manihot esculenta* Crantz (Euphorbiaceae)	Roots: diarrhea and stomachache		Carrageenan-induced paw edema in rats and xylene-induced ear edema in mice	↓ Carrageenan-induced rat paw edema and xylene-induced ear swelling in mice	[146]
*Flueggea virosa* (Roxb. Ex willd.) Royle (Phyllanthaceae)	Root; toothache	Flueggrenes A	N-formylmethionyl-leucyl-phenylalanine (FMLP)/cytochalasin B (CB) activated-human neutrophils	↓ Superoxide anion generation and elastase release	[147]
*Litsea glutinosa* (Lour.) C.B.Rob. (Auraceae)	Leaves and bark; fever, scabies, gastritis pain, cuts, and wounds	Hydroalcoholic extracts of leaves	Carrageenan-induced edema	Carrageenan-induced rat paw edema	[148]
*Merremia tridentata* (L.) Hallier f. (Convolaceae)	Whole plants; sores, antiseptic	Apigenin	LPS-stimulated BV2 microglia	↓ TNF, IL-1β, and IL-6 production ↓ NF-κB activation	[83,124]
		Quercetin, quercitrin	LPS-induced RAW264.7 Cells	↓ NO production, TNF, IL-1β, and IL-6	[82]
*Ocimum tenuiflorum* L. (Lamiaceae)	Leaves and stems; pain and stomachache	Leaf extract	LPS-induced RAW264.7 Cells	↓ Lipopolysaccharide-induced inflammation in RAW 264.7 cells	[149]
*Nauclea orientalis* (L.) L. (Rubiaceae)	Bark; colds, stomachache, and snake bite	Ethyl acetate (EA) and ethyl alcohol (ET) lotus petal extracts	LPS-stimulated human monocyte-derived macrophages	↓ TNF, NF-κB	[150]
*Nelumbo nucifera* Gaertn. (Nelumbonaceae)		Ethanol extract from fruits	Carrageenan-induced paw edema (Wistar male rats)	↓ Carrageenan-induced rat paw edema	[151]
		Leaf extract	293T cells	↑ IL-10 and IL-12 ↓ IL-6, IL-1β, TNF-α, and IFN-γ	[152]
		Neferine	DSS-induce colitis mice model (C57BL/6J male mice)	TNF, IL-1β, and IL-6 ↑ IL-10	[153,154]
			Murine macrophage RAW264.7 cells	IL-6 and TNF production ↑ Peroxisome proliferator-activated receptor (PPARα and PPARγ)	[155]
			LPS-stimulated RAW 264.7 cells	↓ NO release	[156]
*Ochrosia elliptica* Labill. (Apocynaceae)	Bark; malaria	Quercetin-3-O-β-d-glucuronide	LPS-stimulated RAW 264.7 cells and human peripheral blood monocytes	↓ NO production and TNF	[157,158]
*Phyllanthus urinaria* L. (Phyllanthaceae)	Leaves; colds	Corilagin	IB3-1 cells	↓ IL-8 gene expression in TNF-treated IB3-1 cells TNF-alpha-induced secretion of MCP-1	[159]
*Scoparia dulcis* L. (Plantaginaceae)	Whole plants; sores, stomachache, skin lesions, wounds, and cuts	Ethanol extract	Carrageenan-induced paw edema	↓ COX-2, NO, TNF and IL1-β	[160]
		Betulinic acid		↓ TNF, IL-1β, COX-2, and NO production	
*Terminalia catappa* L. (Combretaceae)	Bark, young green fruit; sore throat and thrush	Ethanol extract of leaves	12-O-TPA-induced edema (Male ICR mice)	↓ Myeloperoxidase (MPO) activity	[161]
		Ursolic acid and 2α,3β,23-trihydroxyurs-12-en-28-oic acid		↓ TPA-induced ear edema and inhibited MPO activity	
*Terminalia muelleri* Benth. (Combretaceae)	Leaves: Scabies, sores, skin lesions, wounds, and cuts	Polyphenol-rich fraction	Carrageenan-induced paw edema model	↓ PGE2 and IL-6, IL-1β and TNF, IL-1β	[162]
*Uromyrtus metrosideros* (F.M.Bailey) A.J.Scott (Myrtaceae)	Leaves and branches	(4S)-α-terpineol 8-O-β-D-(6-O-galloyl) glucopyranoside	Human PBMCs stimulated by P/I or anti-CD3/anti-CD28	↓ IFN-γ, IL-17A and IL-8	[64]
*Vitex trifolia* L. (Labiatae)	Leaves; inflammatory ailments	Aqueous extract	LPS-induced RAW 264.7 cells	↓ COX-2, NF-κB, L-1 β and caspase-3	[163]
		Galloyl-lawsoniaside		↓IFN-γ and IL-17A	
*Lophostemon suaveolens* (Soland. ex Gaertn.) Peter G.Wilson and J.T.Waterh. (Myrtaceae)		N-hexane and dichloromethane extracts from leaves	LPS-stimulated RAW264.7 cells	↓ NO production ↓ PGE2 in 3T3 murine fibroblasts	[164]
		Betulinic acid	LPS-induced lung inflammation in Sprague-Dawley rats	Inhibit cell recruitment, ↓ TNF, NO, and TGF-β1 expression ↓ MDA production	[165]
*Tasmannia lanceolate* (Poir.) A.C.Sm. (Winteraceae)	Berries, bark, and leaves	Polyphenolic-rich extracts (gallic acid, 2-hydroxybenzoic acid and chlorogenic acid were identified through their MS/MS spectra)	LPS-stimulated RAW264.7 cells	↓ iNOS and COX-2 ↓ NO and prostaglandin E2 (PGE2) levels	[166]

↓, Downregulated; ↑, upregulated: TNF, tumor necrosis factor; ROS, reactive oxygen species; IL, interleukin; TNF, tumor necrosis factor; IFN-γ, interferon-gamma; NF-kB, nuclear factor kappa beta; MAPk, mitogen-activated protein kinase; NO, nitric oxide; PGE2, prostaglandin 2; COX-1, cyclooxygenase-1; COX-2, cyclooxygenase-2; IL, interleukin; LPS, lipopolysaccharide; NO, nitric oxide; DSS, dextran sulfate sodium; RAW, Ralph and William’s cell line; MPO, myeloperoxidase; DMSO, dimethyl sulfoxide; MMP, matrix metalloproteinase; MDA, malondialdehyde.

**Figure 1 pharmaceuticals-17-00819-f001:**
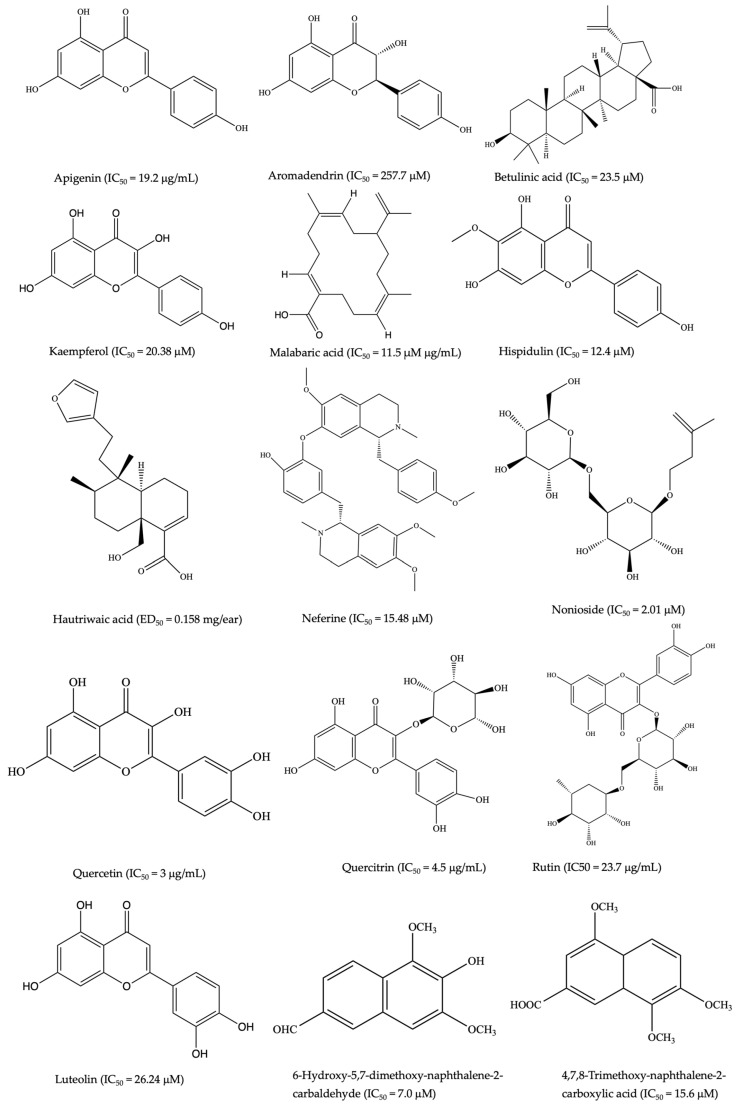
Selected common chemical structures of plant-derived anti-inflammatory SMs (the half-maximal inhibitory concentration (IC50) measures a small molecule’s efficacy by indicating the amount needed to inhibit inflammation by 50%, thus reflecting its potency in treating inflammatory-related disorders) [167].

## 3. Helminth-Derived Anti-Inflammatory Compounds

Several studies have highlighted the therapeutic significance of helminths and helminth-derived excretory/secretory products (ESP) [168] in modulating host immune responses and protecting against inflammatory conditions (Table 2). Helminth-induced immune modulation is multifactorial and often involves the potent stimulation of Th2 responses in conjunction with IL10- and transforming growth factor-β (TGFβ)-dependent immune regulation [169,170,171] and decreased type 1 and 17 T helper (Th1/Th17) inflammation [172,173,174,175]. Helminths are also potent drivers of regulatory cell responses, including Foxp3+ regulatory T cells (Tregs) [171,175,176], IL10-producing type 1 regulatory T (Tr1) cells [177], type 2 innate lymphoid cells [178], alternatively activated macrophages [179] and regulatory dendritic cells [180,181]. Other studies have reported helminths like *Hymenolepis diminuta*, *Trichinella spiralis*, and *Trichuris suis* offering protection against colitis in 2,4-dinitrobenzene sulfonic acid (DNBS) and TNBS-induced mouse models of colitis [182,183] and randomized human trials [184]. For instance, TNBS-treated mice infected with *S. mansoni* eggs showed decreased Th1-type inflammation and alleviated colon pathology, possibly through IL-4 signaling and increased IL-4 levels [185]. The soluble molecules from *S. mansoni* eggs and hookworm (*Ancylostoma caninum*) ESPs were found to counteract the detrimental effects of INF-γ and IL-12 induced by DSS, inducing the secretion of IL-10 [185]. In a piroxicam-induced IL-10^−/−^ mouse model, infection with *Heligmosomoides polygyrus* inhibited IFN-γ and IL-12 production while promoting the production of Th2 cytokine IL-13 [185]. The succinic acid secreted by *Nippostrongylus brasiliensis* ESPs was found to be involved in initiating an early Th2-type immune response [186], which involves the activation of Th2 cells and the production of cytokines like IL-4, IL-5, IL-10, and IL-13. These cytokines promote B-cell differentiation, antibody production, and the activation of eosinophils, contributing to allergic inflammation and the defense against extracellular pathogens [187].

Similar work on *A. caninum* by Wangchuk et al. [188] showed the protection of mice from colitis by low-molecular-weight metabolites of somatic extracts, ESPs, and the latter when used to treat mice resulted in a significant reduction in inflammatory cytokines such as IL-23, TNF, and IL-1β. Similarly, different concentrations of the hexane-dichloromethane-acetonitrile somatic fraction of *A. caninum* exhibited notable reductions in TNF, IL-1β, IL-6, and monocyte chemoattractant protein-1 (MCP-1) production [188]. Hence, helminth therapy is gaining attention in IBD treatment. For example, preliminary investigations into the positive impacts of helminths on IBD revealed that the oral administration of *Trichuris muris* whipworm eggs notably decreased TNBS-induced colitis in IL-10^−/−^ mice [189]. Similarly, *Trichuris suis* ova showed promise in human trials, inducing remission in 21 out of 29 patients with active Crohn’s disease (CD) [184]. Hence, identifying and characterizing helminth molecules offers a unique opportunity to create nature-inspired, effective, safe, and minimally immunogenic drugs.

Wangchuk et al. [190] identified 54 SMs in the ESP of *Trichuris muris* and *N. brasiliensis*, of which 17 SMs had exhibited pharmacological activities in other studies. Similarly, in the case of *Dipylidium caninum*, 49 SMs were characterized by gas chromatography/mass spectrometry (GC-MS), with succinic acid as the chief constituent of its ESP [191]. The study also highlighted that among the 35 polar metabolites, lactic acid, malic acid, methionine, glycerol, and fructose were previously reported to exhibit anti-inflammatory and proinflammatory activities [191]. Furthermore, Wangchuk et al. [192] carried out the initial metabolomic and lipidomic examination of the infective third-stage filariform larvae of the human hookworm *Necator americanus*, utilizing liquid chromatography–mass spectrometry. The study unveiled 645 SMs, with 495 metabolites exclusive to the somatic tissue extract and 34 found solely in the ESP component. Of these, 45 SMs were identified as polar metabolites; 26 SMs had previously been documented for their anti-inflammatory properties [192]. For example, in a study conducted to investigate the impact of L-arginine on the inflammatory response and casein expression, the findings indicated that arginine mitigated the LPS-induced production of inflammatory markers such as IL-1β, IL-6, TNF, and iNOS [193]. Among the SMs identified, lactic acid, malic acid, methionine, glycerol, and fructose, sourced from various NPs, have demonstrated anti-inflammatory properties [194,195,196,197,198]. However, the immunomodulatory properties of nonprotein SMs secreted or excreted by helminths remain relatively underexplored for medicinal applications [199]. Therefore, based on these promising preliminary results, the somatic tissues and ESPs of helminths could be sources of anti-inflammatory SMs that can be used for treating IBD and other inflammatory conditions. The chemical structure of common anti-inflammatory SMs identified through metabolomic studies from *Trichuris muris* and *Ancylostoma caninum* are shown in Figure 2.

**Table 2 pharmaceuticals-17-00819-t002:** Anti-inflammatory activities of different stages of helminths and their components in various colitis animal models and cell lines.

Helminth Species	SMs/ESP/Somatic Tissue/Larva/Egg/EVs/Protein	Animal Model or Cell Line	Mechanism of Action	Reference
*Ancylostoma caninum*	Low-molecular-weight somatic tissue extract metabolites	TNBS-induced colitis model	↓ TNF, IL-1β, and IL-13	[188]
	Low-molecular-weight excretory-secretory product metabolites	LPS-stimulated PBMC	↓ TNF, IL-1β, and IL-13	
	Dichloromethane-acetonitrile somatic extract		↓ TNF, IL-1β, IL-6 and the chemokine MCP-1	
		DSS-induced colitis model mice	↓ Proinflammatory mediators iNOS, IL-6, and IL-17A ↑ IL-4 and IL-10 levels	[200]
*Ancylostoma ceylanicum*	Crude extracts and excretory-secretory products	DSS-induced colitis model	↓ Th1 and Th17 cytokines ↓ Colonic microscopic ↓ EPO and MPO activity	[201]
*Trichinella spiralis*	53 kDa ES protein	TNBS-induced colitis model	↑ IL-4, IL-13, IL-10, TGF-β, AAM	[202]
			↓ IFN-γ, TNF, IL-6 ↓ inflammatory score	
	Extracellular vesicles	TNBS-induced colitis model	↓ IL 1β, TNF IFN-γ, and IL-17A ↑ IL-10 and TGF-β, IL-4, and IL-13	[203]
*Nippostrongylus brasiliensis*	Extracellular vesicles	Murine small intestinal organoids	↓ IL-6, IL-1β, IFN γ, and IL-17a ↑ IL-10	[204]
*Spirometra erinaceieuropaei*	Extracellular vesicles	LPS-stimulated RAW 264.7 cell	↓ NO production ↓ TNF, IL-1β, and IL-6)	[205]
*Schistosoma mansoni*	Egg	TNBS-induced colitis model	↓ IFN-γ ↑ IL-10 mRNA expression	[172]
	Egg	DSS-induced colitis model.	↑ FoxP3+ T regulatory cells and Th2 cytokines.	[206]
*Trichinella spiralis*	Infective larvae	TNBS-induced colitis model	↓ DAI score and weight	[207]
			↓ IFN-γ ↑ IL-4	
*T. spiralis*	Infective larvae	TNBS-induced colitis model	↑ IL-4, IL-13- induction of a Th2 response.	[182]
			↓ IL-12, IFN-γ, MPO activity	
*Heligmosomoides polygyrus*	Infective larvae	Piroxicam- induced IL-10^–/–^ mice	↓ IL-12 and IFN-γ production ↑ IL-13,	[170]
	Infective larvae	TNBS-induced colitis model	↑ IL-4, IL-5, IL-10, IL-13	[208]
			↓ IL-12p40, IFN-γ ↑ IL-4, IL-5, IL-13, and IL-10 secretion	
	Infective larvae	IL-10^−/−^ mice	↓ IL-17 ↓ IL-4 and IL-10	[209]
	Infective third-stage larvae	TNBS-induced colitis model	↑ IL-4, IL-13, mucosal mast cells, and resistance	[210]
			↓ IFN-γ, TNF	
*Trichuris muris*	Embryonated eggs	Mdr1a^−/−^ mice	↑ IFNγ, TNF ↑ IL-13 and IL-5	[211]
	Infective larvae	IL-10^−/−^ mice	↑ IFN-γ, IL-17	[212]
			↓ IL-13 ↑ IL-13Rα2	
*Necator americanus*	Netrin-domain-containing proteins (prophylactic *Na*-AIP-1)	TNBS-induced colitis model	↓ TNF	[213]

## 4. Anti-Inflammatory Agents in Clinical Trials

Several plant species and their allied small molecules are being examined for their potential in treating IBD, specifically UC and CD. The small molecule epigallocatechin-3-gallate derived from *Camellia sinensis* is being investigated for its safety in patients with mild to moderately active UC during clinical remission and maintenance therapy, with the study currently in phase II. Another anti-inflammatory SM, curcumin, is also being investigated at various clinical phases. In pediatric patients with IBD, a phase I study is aiming to assess the tolerability of curcumin. In patients with UC, a phase I trial is investigated the effectiveness of a combined therapy involving curcumin and 5-ASA compared to that of 5-ASA alone (Table 3).

In contrast, a phase III trial is exploring the impact of combining curcumin with thiopurines to prevent postoperative recurrence. The bioactive compound berberine, isolated from *Coptis chinensis*, is the focus of a phase I study evaluating its safety in patients with UC in clinical remission and undergoing maintenance therapy. Finally, triptolide, reported as a bioactive compound in *Tripterygium wilfordii*, is being studied to assess its impact and safety in inducing remission in CD, comparing its efficacy with that of mesalamine, with ongoing trials in both phase II and III. Overall, studies underscore the potential of natural compounds in treating BD, offering insights into safety, efficacy, and therapeutic strategies at the various stages of clinical development (Table 3).

Following the positive tolerance and absence of side effects observed in the open trial involving patients with active UC and CD, researchers conducted clinical trials primarily using *Trichuris suis*, the pig whipworm, and *Necator americanus*, a hematophagous hookworm [169]. In an open-label trial (phase 1) at the University of Iowa, seven patients with IBD received 2500 TSO, resulting in remission for six patients per the IBD Quality of Life Index [169,214]. Subsequently, a randomized, double-blind, placebo-controlled trial (NCT01433471) involved 54 active UC patients treated with 2500 TSO or placebo every two weeks for 12 weeks. The study showed significant improvement in 43.3% of patients treated with TSO compared to 16.7% of placebo recipients [169,214]. Similarly, a trial with 29 patients with active CD demonstrated a 79.3% decrease in the CD activity index (CDAI) and a 72.4% remission rate after 24 weeks of TSO treatment (Table 3). *N. americanus* was also tested in patients with CD, showing clinical improvement and remission in eight of nine patients after 20 weeks despite mild side effects. Additionally, helminth-derived products, such as P28 S-glutathione transferase (P28SGT), demonstrated promise, reducing disease activity and inflammatory markers in patients with CD in clinical trials (NCT02281916) [169] (Table 3). Considering the reported immunomodulatory properties, helminths show significant potential for treating inflammatory bowel disease (IBD). However, further research is needed to fully understand their mechanisms and ensure their safe and effective application in clinical settings.

## 5. Advances and Recent Approaches in SM Drug Lead Discovery

In drug discovery, including SMs, drug leads demand the physical screening of large chemical libraries for biological targets, which is common but time-consuming and expensive. As such, there have been substantial advancements in techniques for SM drug discovery, including high-throughput screening, structure-based drug design, virtual screening, and the refinement of lead compounds. High-throughput screening (HTS) analyzes over a million compounds biochemically, requiring substantial time and investment [215]. To address this, virtual high-throughput screening was developed as a cost-effective computational method that is extensively applied in early drug discovery. It aims to identify novel, active small molecules by searching vast compound libraries, supporting the goals of HTS while reducing costs by evaluating only selected compounds for pharmacological activity [216,217].

Furthermore, molecular docking has proved to be a crucial approach in drug discovery in predicting the interaction patterns between proteins and small molecules and indicating the presence of bioactive compounds in natural products. For instance, GC-MS-identified bioactive ingredients and molecular docking against key targets revealed 3,5-dehydro-6-methoxy, ethyl iso-allocholate cholest-22-ene-21-ol, alpha-cadinol, and pivalate of *Phyllanthus nivosus* as promising compounds for UC drug development [218]. Computational methods, particularly in silico discovery, enhance traditional drug development, ensuring sustainable and cost-effective drug discovery with increased efficacy [219]. Modern techniques like pharmacophore modeling are also crucial for virtual screening, utilizing advanced compound databases and computing power to find small molecules of lead compounds [17].

To expedite the discovery of bioactive NPs in extracts, metabolomics data have been subjected to chemometric methods like multivariate data analysis, which correlate measured activity with nuclear magnetic resonance (NMR) and MS spectra signals, facilitating the tracking of active compounds in complex mixtures without additional bioassays [168,220]. Recent advancements in analytical technologies, particularly higher-field NMR instruments and probe technology, have allowed for precise structure determination of NPs even from limited quantities (<10 µg) [221]. Microcrystal electron diffraction, a cryo-electron-microscopy-based technique, is being increasingly applied for unambiguous structure determination of SMs in NP research [222]. Bioactivity-guided fractionation techniques with NMR-based methods have recently been utilized for screening, identifying, and isolating anti-inflammatory bioactive compounds from natural products. For instance, a methanolic extract of *Uraria crinite* (L.) roots was screened to isolate the immunomodulatory isoflavone genistein. This compound exhibited immunomodulatory activity against producing proinflammatory cytokines (IL-6 and TNF) [223].

## 6. Challenges and Future Directions

Exploring the small anti-inflammatory molecules derived from remedial plants and helminths represents a promising frontier in pharmacopoeia research. These naturally sourced compounds often exhibit unique mechanisms of action, lower toxicity, and fewer side effects. However, challenges exist in identifying and isolating bioactive compounds from diverse natural products. The use of HTS, computational approaches like molecular docking and virtual screening, and integrating artificial intelligence (AI) and machine learning (ML) algorithms can enhance the accuracy and efficiency of identifying promising candidates from natural compound libraries. It is crucial to understand the molecular mechanisms through which these natural compounds exert their anti-inflammatory effects. Research should focus on investigating how these compounds interact with key inflammatory mediators. Advanced techniques such as clustered regularly interspaced short palindromic repeats (CRISPR)/CRISPR-associated protein 9 (Cas9) gene editing and RNA sequencing can provide better insights into these interactions. Additionally, addressing challenges related to bioavailability and unfavorable pharmacokinetic profiles is essential. Techniques like nanoencapsulation, liposomal delivery, and phytosome technology should be incorporated to improve small anti-inflammatory molecules’ absorption, stability, and targeted delivery. Research has shown that helminths are therapeutic in treating inflammatory disorders due to their mechanisms evolved for modulating host immune responses. Further research should explore isolating and characterizing the small molecules from helminths to assess their therapeutic potential in inflammatory diseases. Techniques like proteomics and metabolomics can be instrumental in identifying bioactive helminth-derived compounds. Continued interdisciplinary research and collaboration will unlock the full therapeutic potential of these natural compounds.

## 7. Conclusions

In conclusion, our comprehensive analysis of the anti-inflammatory properties inherent in natural products, encompassing both crude extracts and isolated SMs, underscores their remarkable capacity to modulate a spectrum of inflammatory pathways. The anti-inflammatory efficacy of natural products is manifested by inhibiting key inflammatory mediators such as NO, Cox-2, and proinflammatory cytokines, alongside the stimulation of anti-inflammatory cytokine production. While certain reported extracts or SMs operate through singular or dual mechanisms, others exhibit a more diverse array of actions. Furthermore, emerging research highlights the therapeutic promise of helminths and their secretory products (ESPs) in coordinating host immune responses and alleviating inflammatory maladies. Helminth-induced immune modulation fosters a milieu conducive to Th2-, IL10-, and TGFβ-dependent immune regulation, effectively attenuating Th1/Th17 inflammatory responses. Several helminth species, including *Schistosoma mansoni, Hymenolepis diminuta*, *Trichinella spiralis*, and *Trichuris suis*, demonstrate significant protective effects against colitis in preclinical models and human trials, highlighting their potential as therapeutic agents.

In light of these findings, NPs remain a fertile ground for identifying and discovering diverse structures of anti-inflammatory SMs. These structures can be either directly developed or serve as initial frameworks or scaffolds for further optimization into innovative anti-inflammatory drugs. Despite the challenges associated with drug development, such as high attrition rates and obstacles related to accessibility, sustainable supply, and intellectual property constraints, we remain optimistic that ongoing scientific and technological advancements will establish a robust foundation for NP-based drug discovery and harness the vast potential of nature’s pharmacopeia.

## Figures and Tables

**Figure 2 pharmaceuticals-17-00819-f002:**
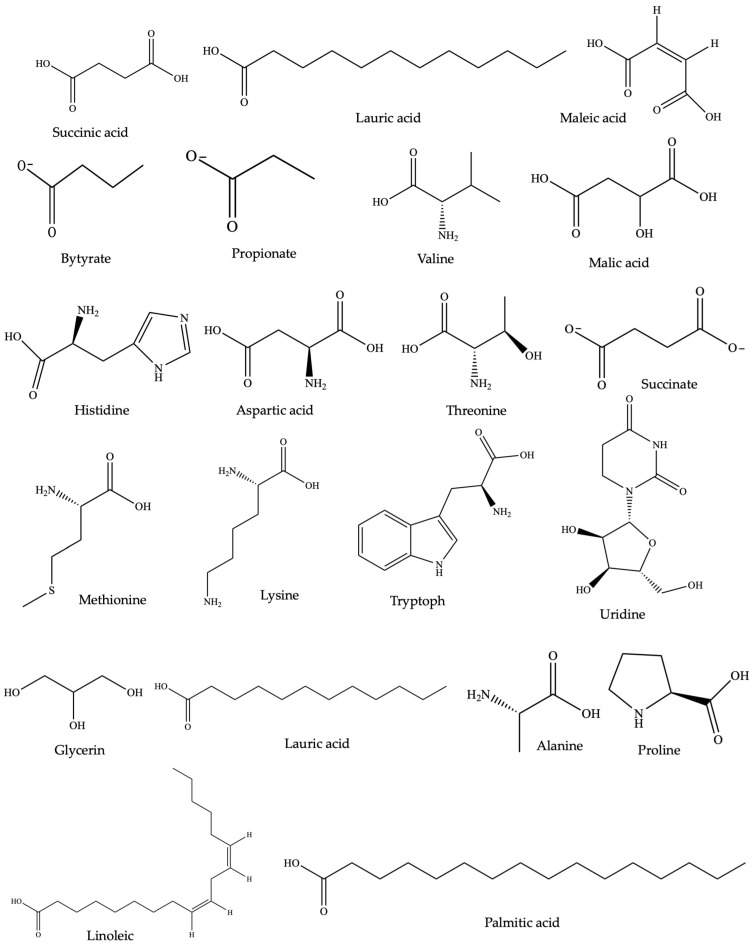
Chemical structure of anti-inflammatory SMs identified through metabolomic studies of *Trichuris muris* and *Ancylostoma caninum* (common to both).

**Table 3 pharmaceuticals-17-00819-t003:** Nature-derived anti-inflammatory compounds and helminth products in clinical trials for treating inflammatory bowel diseases.

Plant Species	Natural Therapeutics	Diseases	Objective	Clinical Phase	Clinical Trial Number
Plant-derived compounds/drugs					
*C. sinensis* L.	Epigallocatechin3-gallate (Polyphenon E^®^)	Mild to moderately active UC	Assess safety of an oral dose of green tea extract as initial evidence to substantiate its effectiveness in UC (oral administration of placebo tablet)	Phase II	NCTT00718094
*C. longa* L.	Curcumin	Both UC and CD	Assess the tolerability of curcumin in pediatric patients with IBD (double-blinded placebo-controlled study)	Phase I	NCT00889161
*C. longa* L.	Curcumin	UC	Assess effectiveness of a combined therapy involving curcumin and 5ASA compared to that of 5ASA alone in patients with mild to moderate UC (randomized, double-blind, placebo-controlled study)	Phase III	NCT01320436
*Coptis chinensis* Franch	Berberine (berberine chloride)	UC	Examine the safety profile of berberine in individuals with UC who are in clinical remission and are simultaneously undergoing maintenance therapy with mesalamine (oral administration of placebo)	Phase I	NCT02365480
*Tripterygium wilfordii* Hook. F.	Triptolide (Tripterygium glycosides)	CD	Evaluate the efficacy of combining Tripterygium glycosides with enteral nutrition in inducing remission in patients with Crohn’s disease, comparing outcomes with those receiving either Tripterygium glycosides alone or enteral nutrition alone	Not applicable	NCT01820247
Helminths products/drugs					
*Trichuris suis*	*Trichuris suis* ova (CNDO 201)	CD	Evaluate the safety and tolerability of single oral doses of CNDO 201 (double-blind, placebo-controlled)	Phase I	NCT01434693
*Trichuris suis*	*Trichuris suis* ova (CNDO 201)	UC	Investigate the immune response activated in the human gastrointestinal tract (double-blind, placebo-controlled oral inoculation)	Phase I and II	NCT01433471
*Trichuris suis*	*Trichuris suis* ova (TSO)	CD	Compare the efficacy of three doses of oral TSO suspension versus a placebo in inducing remission	Phase II	NCT01279577
*Schistosoma mansoni*	P28GST (protein 28 Kd glutathion S Transferase)	CD	Assess the effectiveness in regulating immunologic and inflammatory markers in blood and tissue, as well as determining the occurrence of clinical recurrence using the disease activity index	Phase II	NCT02281916

The sources were acquired from www.clinicaltrials.gov (Accessed: 16 February 2024).

## Data Availability

Data sharing is not applicable.

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
