# Peer review of "Pharmaceutical Potential of Remedial Plants and Helminths for Treating Inflammatory Bowel Disease"

_pharmaceuticals, 2024, doi:10.3390/ph17070819_

Round 1

Reviewer 1 Report

Comments and Suggestions for Authors

The review entitled “Anti-inflammatory small molecules of remedial plants and gastrointestinal parasites” by Tenzin Jamtsho et al., looks very clear and interesting as the authors highlighted the important natural products with potential application in inflammation.  Overall the review is well written with adequate information for dissemination.

I would like to suggest the following prior to acceptance for publication.

1)     The title needs to be changed/modified.

2)     Nice to mention the class of compounds derived in general with SAR for being effective in inflammatory diseases.

3)     Better to mention the name of the molecules/metabolites derived from the Helminth in Table 2.

4)     Used the word anti-inflammatory molecules instead of anti-inflammatories throughout the text.

5)     In Table 3, separate the Nature derived anti-inflammatories in clinical trials and existing approved natural anti-inflammatories drugs

Comments on the Quality of English Language

English language is fine

Reviewer 2 Report

Comments and Suggestions for Authors

Dear authors,

Please, revise the manuscript considering the following comments:

1 Introduction

Include the recent advances on the administration of bioactive molecules of plants and helminths, and their efficacy in comparison to synthetic drugs.

L- 41 – Include the side effects reported after NSAID administration.

L55-57 – Any idea why/the origin of these effects? Any molecule in specific of these medications induce these side effects?

Topic 2

L90 – Define ‘bioactive compounds’

L94-120 – Which molecules present in these plants are anti-inflammatory?

L168-172 – Explain the role of nitric oxide, IL-6, TN-alpha, IL-1beta and others. On the inflammation and how the activity of these proteins are reduced after administration of anti-inflammatory molecules.

Table 1- Include the recent reviews (preferably 5 years to now) preferably.

Table 1- There are lines (like the 3 lasts) that do not include the type or which molecule was prominent in plant extracts used.

Figure 1 – Define what IC50 is and how it indicates the molecule potential?

213 – What is the ‘Th2-type immune response’?

Topic 4 - How these phytochemicals were efficient in comparison to synthetic drugs?

Table 3 - Were these trials conducted with controls?  Also, the table does not include the trials with drugs, but phytochemicals.

Additional  observation: there are many biomarkers not defined and importance not mentioned. Avoid dumping results 

Topic 5 - Which future directions authors propose? Are there methods to valorize bioactives indigenous to helminths? Is there a trend on using that against inflammation?

Reviewer 3 Report

Comments and Suggestions for Authors

The current manuscript is good and well written. The authors' review summarizes the anti-inflammatory properties of indigenous medicinal plants and tropical helminths. Additionally, the review explores recent advances and approaches in discovering small molecules with anti-inflammatory properties to address the complexities associated with IBD. As noted, the current title of the manuscript focuses on medicinal plants, while the journal is named "Pharmaceuticals." I recommend the authors update the text to better align with the journal's scope.

Comments:

1) The abstract does not provide details on the literature search methodology, which is important to assess the completeness and rigor of the review. Some clarification on the search strategy and inclusion/exclusion criteria would strengthen the review. However, ensuring a comprehensive and systematic literature search is crucial for the credibility and completeness of the review.

2) Keywords should be check according to MeSH. 

3) Discuss the key challenges and future research directions in the development of natural-product-based anti-inflammatory therapies. I recommend write as new subtitle at the end of text body as discussion.

4) Table 1, Page 14 of 40: Please check the Botanical name and family  the reference 122 not supported.

5) Table 1, Page 14 of 40: ethanol Change to "Ethanol"

6) The current articles under below should be recommend to use as new update of bioactive compound:  

DOI: 10.1016/B978-0-323-91740-7.00019-0

Comments on the Quality of English Language

The English is suitable. 

Round 2

Reviewer 2 Report

Comments and Suggestions for Authors

Dear authors,

Before publication of manuscript I recommend rewrite the abstract. Consider addressing at most the novelties provided by this review, the current trends and future directions. 

For instance, from line 10 to 24, there is just contextualization and there is anything related to the main topics found on this review.

Author Response

We sincerely appreciate your valuable comments and suggestions. Your insights have been instrumental in enhancing our manuscript and ensuring it aligns with the journal's scope. Given the significance of your feedback, we have completely rewritten the abstract. Thank you once again.
